# A COVID-19 Overview and Potential Applications of Cell Therapy

**Diana Aparecida Dias Câmara** [1,*], **Allan Saj Porcacchia** [2], **Nelson Foresto Lizier** [1] and **Paulo Luiz De-Sá-Júnior** [3,*]

1   Nicell-Pesquisa e Desenvolvimento Científico Ltda., São Paulo 04063-005, SP, Brazil; nelson.lizier@nicell.bio
2   Departamento de Psicobiologia, Universidade Federal de São Paulo (Unifesp), São Paulo 04024-002, SP, Brazil; allansaj.7@gmail.com
3   Paulo Luiz de Sá Júnior Eireli, São Paulo 03303-000, SP, Brazil
*   Correspondence: diana.camara@nicell.bio (D.A.D.C.); paulsaj2001@yahoo.com.br (P.L.D.-S.-J.); Tel.: +55-11-2667-4580 (D.A.D.C.)

**Abstract:** The COVID-19 pandemic has already reaped thousands of lives, although many scientific studies already showed the possibility of this scenario. Currently, further attention is provided to patients depicting comorbidities such as respiratory or immunocompromised diseases, hypertension, and diabetes, as these individuals show a worse prognosis. Cell therapies using stem cells and/or defense cells, combined or not with traditional treatment, could be an outstanding strategy for COVID-19 management since these treatments can act by modulating the immune system, reducing proliferation, and favoring the complete elimination of the virus. In this review, we highlight the main molecular characteristics of this novel coronavirus, as well as the main pathognomonic signs of COVID-19. Furthermore, possible cell therapies are pointed out to show alternative treatments against COVID-19 and its sequels.

**Keywords:** COVID-19; SARS-CoV-2; hematologic disease; cell therapy; stem cell





## 1. Introduction

A new pandemic that would reap thousands of lives was already expected in recent decades [1]. Several factors are related to the emergence of this new pandemic disease, all of them related to anthropic actions, such as, but not limited to, migration, the expansion of agricultural areas, and deforestation. All these factors act together facilitate the contact of undiscovered or even new viruses with humans and enable the emergence of new zoonoses. In December 2019, a total of 41 cases of severe pneumonia of unknown origin were confirmed in the city of Wuhan, Hubei province, central China, giving rise to the most serious pandemic of the century [2].

Coronaviruses were described for the first time in 1966 by Tyrell and Bynoe, who cultivated viruses from patients with common colds. The term coronavirus alludes to their morphology, which resembles spherical virions with a core–shell and surface projections like a solar corona. They were then termed coronaviruses (Latin: *corona* = crown) [3]. The causal agent of the syndrome was primarily named novel coronavirus (2019-nCoV). Afterward, based in their close genetic relationship with severe acute respiratory syndrome coronavirus (SARS-CoV), the new virus was renamed by the International Committee on Taxonomy of Viruses (ICTV) as SARS-CoV-2 [4]. Coronavirus disease 19 (COVID-19) was the name given to this new disease, and the World Health Organization (WHO) confirmed the pandemic and the danger of this disease. COVID-19 has resulted in the death of many people, and despite that, some governments in different countries neglected isolation and masking measures, the only effective non-pharmacologic procedure in preserving human lives to date.

COVID-19 symptoms can vary widely, ranging from milder to extremely severe manifestations. Initial cases reported lower respiratory tract infection-related symptoms [5], which are symptoms similar to those previously observed in two other coronavirus diseases, severe acute respiratory syndrome (SARS), and Middle East respiratory syndrome (MERS) [6]. Headache, dizziness, generalized weakness, vomiting, and diarrhea have also been observed and linked to the disease [7]. Severe symptoms, such as lung injury, breathing difficulty, and coughing up blood have been described at all ages. Particularly in older patients or in those affected by multimorbidity, the virus is more likely to cause severe interstitial pneumonia, acute respiratory distress syndrome (ARDS), and subsequent multiorgan failure, which are clinical manifestations responsible for respiratory failure and high death rates. Typically, COVID-19 patients display a variable extent of dyspnea and radiological signs [5]. Skin manifestations, such as pseudo-chilblains and vesicular lesions, hearth injury, hepatic and kidney failure are reported in patients from different hospitals following the viral infection, which has caused a rise in uncertainties regarding the physio-pathogenic mechanisms of the SARS-CoV-2 infection [6,8].

The keystone for managing possible or confirmed cases is early triage and isolation [9]. Therapeutic approaches for patients with acute respiratory failure are performed by using antiviral agents associated or not with anti-inflammatory drugs. Is spite of that, there is no specific drug or therapy to combat SARS-CoV-2 so far, and the search for these specific agents has been an urgent and necessary task. Currently, only mass vaccination against COVID-19 can reduce the number of new infections and deaths and improve the pandemic scenario.

In this review, we describe the characteristics of SARS-CoV-2, including pathophysiological mechanisms known so far, and cell therapies that are promising in the management of patients with COVID-19 and its sequels. These therapies might be used alone or associated with other traditional treatments previously standardized, and may be important in treating other viral infections that may arise in the future. Once core protocols are established, it is possible to customize the treatment for new specific targets, allowing for rapid development, as observed in the development of COVID-19 vaccines.

## 2. SARS-CoV-2 Structure

In phylogenetic terms, SARS-CoV diverges from other coronaviruses formerly described [10]. Its genome is composed of a reasonably conserved region encoding an RNA-dependent RNA polymerase and a variable region containing open reading frames (ORFs) encoding sequences for the viral structural proteins small envelope (E) glycoprotein, membrane (M) glycoprotein, nucleocapsid (N) protein, and spike (S) glycoprotein, which are necessary to generate a viable and fully structured viral particle [11,12]. Certain coronaviruses can form an infectious virion without the full ensemble of structural proteins. This suggests that supplementary proteins with overlapping compensatory functions may be encoded by these viruses and/or that some structural proteins are not crucial to the capacity of infection and to the viral cycle [13,14].

Among 76–109 amino acids compose the membrane E protein of coronaviruses, which ranges from 8.4 to 12 kDa in size [15] and is the smallest of the major structural proteins [14]. A short and hydrophilic amino terminus consisting of 7–12 amino acids composes the primary and secondary structures of E protein, followed by a large hydrophobic transmembrane domain composed by 25 amino acids. It ends with an extended and hydrophilic carboxyl terminus, constituting most of the protein. During the replication cycle, infected cells express high quantities of E protein [14].

The structural protein present in a greater quantity is the M protein. It is responsible for the shape of the viral envelope and for the virion organization, assembling the other main structural proteins by homotypic interactions. [16]. Taken together, M and E glycoproteins compose the viral envelope, and their interaction is responsible for the production and shedding of virus-like particles (VLPs) [16]. The interaction of S with M is needed to trap S in the endoplasmic reticulum–Golgi intermediate compartment (ERGIC)/Golgi

tcomplex [11,16] and the N protein steadies the nucleocapsid (N protein–RNA complex), as well as the internal core of virions [14].

The S glycoprotein (known as S-protein) mediates the attachment of the virus to surface cell receptors. This molecule is responsible for the fusion between the virus and the host cell membrane, allowing the viral infection [1]. The subunits S1 and S2 compose the coronavirus S protein. Receptor binding is mediated by S1, while the membrane fusion process is made by S2. Coronaviruses typically possess two domains within S1 capable of binding to host receptors: an amino N-terminal domain and a carboxy C-terminal domain. The latter one recognizes protein receptors for SARS-CoV and MERS-CoV [17]. Although these individual domains have been structurally characterized, the organization of the complete S-protein remains to be elucidated.

### 3. Spike Glycoprotein and ACE2 Receptor Interaction

Previous studies with SARS-CoV highlighted the importance of the angiotensin I converting enzyme 2 (ACE2) receptor during this virus infection [18]. The envelope of coronaviruses contains a homotrimeric S-protein, which allows viral particles to bind on the cell membrane [19]. This interaction between S-protein and ACE2 is essential to the SARS-CoV entrance inside human cells [19], and the same has been observed with SARS-CoV-2 [20]. While the S1 subunit of S-protein and ACE2 dissociate, the S2 subunit turns into a more stable state, which permits the membrane fusion [21]. Computer modeling of interactions between SARS-CoV-2 S-protein and ACE2 receptor made by Lan and colleagues (Figure 1) showed that other unknown residues are also involved among the two molecules [19], a feature that corroborates the findings of two other bio-structural studies on SARS-CoV-2 [22]. The computer modeling also emphasized the existence of thirteen hydrogen bonds and two salt bridges, both hydrophilic interactions, between SARS-CoV-2 S-protein and ACE2 receptor, which is similar to the hydrophobicity affinity seem between SARS-CoV S-protein and ACE2 receptor [19].

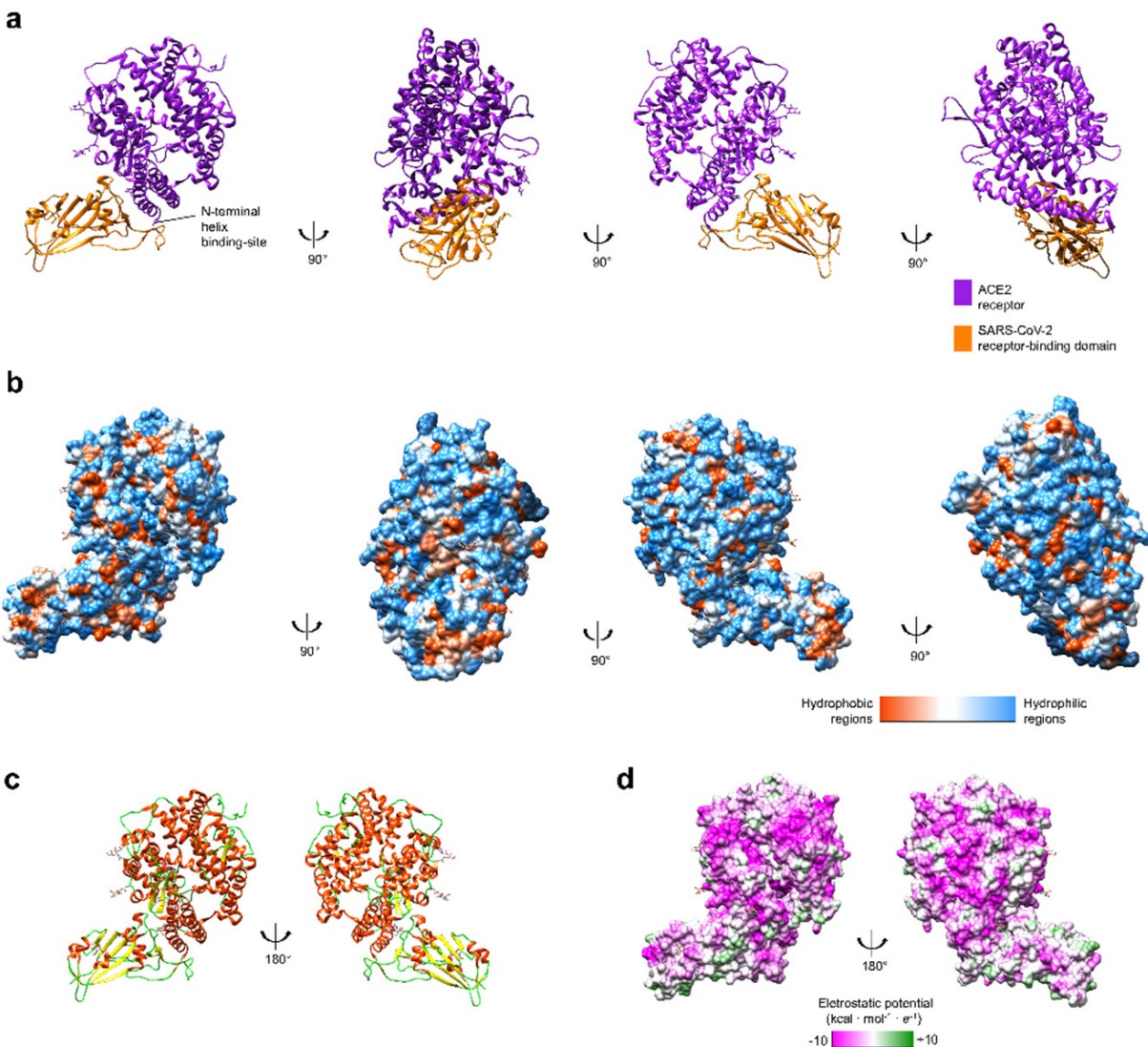

**Figure 1.** (**a**) Tridimensional structures of spike (S)-protein and ACE2 receptor structure (UniProt structure identifier 6M0J [19]). (**b**) Hydrophobicity surface colored following the Hessa and von Heijne hydropathic scale thresholds (dark orange most hydrophobic; white 0; aquamarine most hydrophilic) as described by Hessa et al. [23]. (**c**) The secondary structure is colored in red (helix), light green (random coil), and yellow (strand). (**d**) Electrostatic surface colored according to Coulombic electrostatic potential, e = 4r, thresholds ±5 kcal·mol$^{-1}$·e$^{-1}$ at 298 K. Molecular graphics and analyses performed using the UCSF Chimera package software [24].

The activation of coronavirus' S-protein also depends on another protease to cross the cell membrane. As it was seen in SARS-CoV, transmembrane protease serine-type 2 (TMPRSS2) interacts with S-protein, cleaving it on the cell surface [25]. This induces cell–virus fusion and enables the infection of the human cell [26]. The cleavage mechanisms of TMPRSS2 (Figure 2) were also observed in other viral infections. In this scenario, new therapies that are aimed at active-site inhibitors of TMPRSS2 may be relevant to treat COVID-19 patients and even to be used preventively by health workers due to its potential protective effect, as long as there are no adverse effects. Randomized clinical trials are necessary to evaluate this.

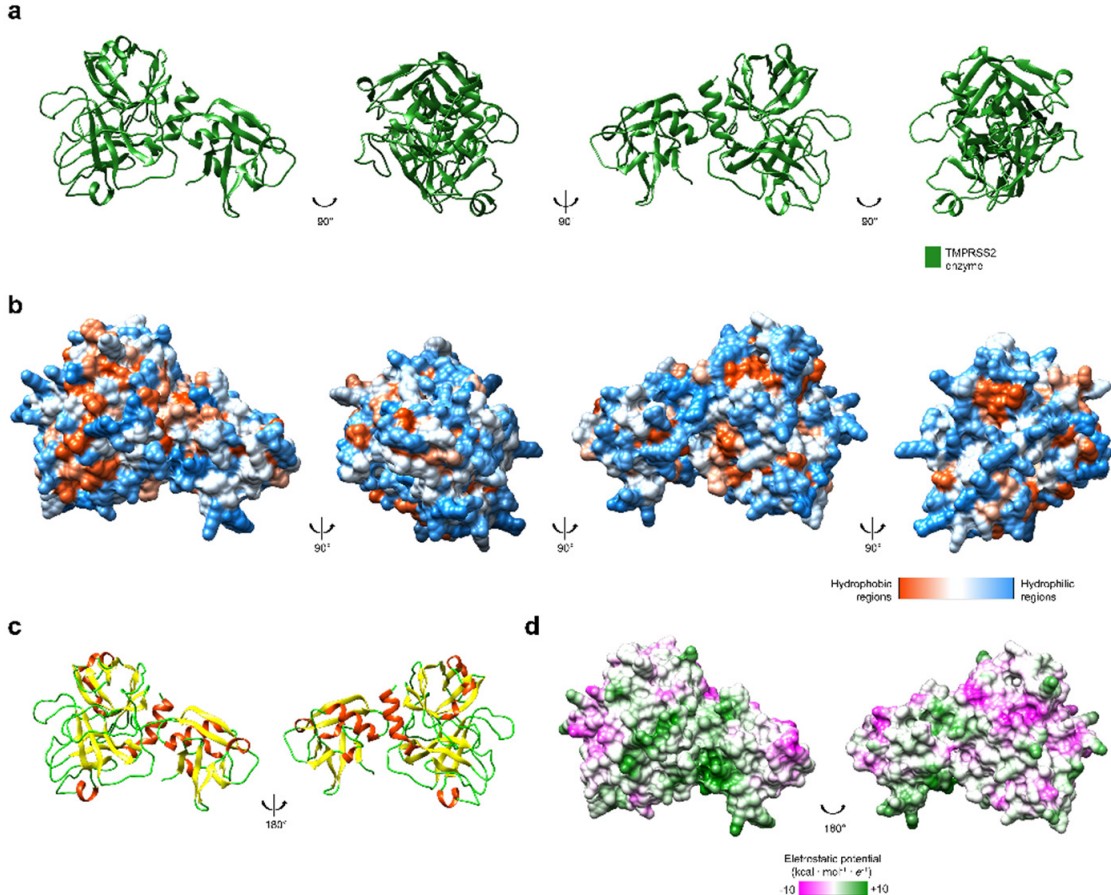

**Figure 2.** (**a**) Tridimensional structure of TMPRSS2 (UniProt structure identifier O15393 [27]). (**b**) Hydrophobicity surface colored following the Hessa and von Heijne hydropathic scale thresholds (dark orange most hydrophobic; white 0; aquamarine most hydrophilic) as described by Hessa et al. [23]. (**c**) The secondary structure is colored in red (helix), light green (random coil), and yellow (strand). (**d**) Electrostatic surface colored according to Coulombic electrostatic potential, e = 4r, thresholds $\pm 5$ kcal·mol$^{-1}$·e$^{-1-}$ at 298 K. Molecular graphics and analyses performed using the UCSF Chimera package software [24].

Jin and colleagues showed the relevance of another enzyme during the SARS-CoV-2 viral replication: Mpro [28]. This protease digests the two overlapping polyproteins pp1a and pp1ab, which are enrolled in the viral transcription and replication [29]. Thus, Mpro is essential to the viral cycle. If humans do not have Mpro close homologues, it rises as a possible drug therapeutic target aiming to treat COVID-19 [28].

## 4. Hematologic Disease

Several studies have shown that SARS-CoV-2 infects host cells by recognizing ACE2 receptor located on the cell membrane [30]. ACE2 receptor is expressed in cells of several human tissues and organs: capillary epithelium, alveolar type 2 (AT2) epithelial cells, alveolar monocytes, macrophages, heart, lung, liver, kidney, intestine, gut [31], endothelial cells, and hematopoietic stem cells [30].

Although COVID-19 is considered a respiratory tract infection, it is, in fact, a systemic disease, as it includes pleiotropic viral dissemination leading to damages and alterations in many systems including immune, cardiovascular, gastrointestinal, hematopoietic, skin, and even neural systems [32].

An intrinsic characteristic presented by many COVID-19 patients is that, although they showed low levels of oxygen in the blood, part of their lung mechanics was still well-preserved [33], which means that breathing movement was not directly affected. Due to this, some studies suggest the involvement of pulmonary vascular vessels in the

pathophysiology of the disease. In these cases, oxygen uptake in COVID-19 pneumonia would not be caused only by edema fluid in alveoli but by constricted blood vessels in the lungs [34]. Furthermore, a high rate of venous thromboembolism has been observed in COVID-19 patients in a critical state, even in those that do not have classic risk factors for this symptom [35]. Patients that show a severe or critical state of the disease exhibit a high incidence of aberrant coagulation mechanisms [30]. Poor and colleagues suggested that pulmonary endothelial dysfunction, resulting from diffuse pulmonary microthrombi and vascular dilatation in different regions of the lung, is the primary cause of respiratory failure in COVID-19 patients [35]. Prolonged prothrombin time, and high levels of fibrinogen and D-dimer (a fibrin breakdown product) are some factors that contribute to the hypercoagulable state detected in COVID-19 patients [30].

Severe inflammatory status is also related to alterations in the coagulation cascade. The intense release of cytokines during the illness, known as a cytokine storm, plays an important role in this process. Some of the cytokines in high plasma concentrations in COVID-19 patients were interleukin-6 (IL-6), interleukin-1β (IL-1β), tumor necrosis factor-α (TNF-α), and interferon γ-inducible protein (IP-10) [36]. Together, the endothelial cell injury, pulmonary viral attack, peripheral endothelial injury, and specific proinflammatory cytokines, such as IL-6, may induce the intense activation of coagulation, leading to the hypercoagulation state in these patients. The dysfunctional coagulation may also lead to an altered and aggressive immune response [30], which makes this process a no-end feedback loop. The increase in procoagulant factors, including fibrinogen and D-dimer, both characteristics of coagulopathy, have been related to the higher mortality of COVID-19 patients [37].

COVID-19 progression is associated with low levels of lymphocyte (lymphopenia) and higher levels of neutrophils, ferritin, IL-6, and D-dimer. Xiang and collaborators [38] reported in peripheral blood mononuclear cells the upregulation of pathways related to apoptosis, autophagy, and p53 protein. More than 80% of COVID-19 patients showed lymphopenia right in the beginning of the illness [39], which is associated with T cells' decrease, mainly CD8+ T cells, and is positively correlated with the aggressivity of the illness [40]. The COVID-19 infection of the endothelium and the immune-mediated responses cause the recruitment of immune cells and can generate a general endothelial dysfunction associated with apoptosis [41].

A study of post-mortem COVID-19 patients' histological analysis showed the accumulation of inflammatory characteristics in endothelium cells, lymphocytic endotheliitis, and apoptotic bodies in different tissues and organs, including the lungs and heart. According to the authors, these features raised the possibility of SARS-CoV-2 promotion of endotheliitis in many organs, which explains a systemic dysregulation in circulatory and microcirculatory functions [41], as there was thrombosis and microthrombosis in several vessels, with a wide variety of cardiovascular complications.

The hypercoagulable state of COVID-19 patients led the International Society on Thrombosis and Hemostasis to propose the administration of anticoagulation drugs to COVID-19 patients in hospitals to reduce mortality [40]. The use of low molecular weight or unfractionated types of heparins were preferred over direct oral anticoagulants, as they may have pharmacological interactions with parallel antivirals and antibiotics [42]. Another possible therapeutic approach that still lacks data to be elucidated is the use of mesenchymal stem cells (MSCs), as they have the capability to migrate to inflammatory sites in the organism and anti-fibrotic, anti-inflammatory, and immunomodulatory effects [43].

## 5. Stem Cell Treatment

The immunomodulatory effect of MSCs can stimulate the metabolism, not only through the secretion of chemokines, growth factors, and cytokines, but also through the production of secretomes and proteomes. These features allow the application of these allogeneic/heterologous MSCs in cell therapies, helping mainly in autoimmune diseases and organ transplantation, as well as regulating angiogenesis and apoptosis. [44]. Due

to the low expression of human leukocyte antigens (HLA) and co-stimulatory molecules under no stimulatory circumstances, MSCs are naturally immunoprivileged [45]. Another important characteristic of these cells refers to the immunosuppressive actions activated significantly under the inflammatory stimulus, in the presence of interferon-δ (IFN-δ), TNF-α, and IL-1β [46,47].

In vivo studies have shown that after the injection of MSC, there is a differentiation of macrophages (M2) that show anti-inflammatory and phagocytic action [48]. The in vitro and in vivo immunosuppressive potential of MSCs has been demonstrated in a wide variety of pathologies and conditions, such as graft versus host disease (GvHD), autoimmune diseases, and organ transplants (skin, cornea, liver, kidney). Through various clinical trials, the safety and efficacy of MSCs [49] to treat immune-mediated inflammatory diseases [50], can be documented. Furthermore, MSCs could interact with many kinds of immune cells, including B cells, T cells, dendritic cells (DCs), natural killer (NK) cells, neutrophil, and macrophages, thereby helping modulate inflammatory responses and balance immune profiles [51].

SARS-CoV-2 can stimulate a high increase in cytokines in the lung, such as IL-2, IL-6, IL-7, G-SCF, IP-10, MCP1, MIP1A, and TNF-α, followed by edema, fibrosis, dysfunction of the air exchange, acute respiratory distress syndrome, severe cardiac injury, and the secondary infection [52]. These effects might be remedied by MSCs and their immunomodulatory and therapeutic action.

Leng and collaborators demonstrated that intravenous MSCs infusion could reduce the overactivation of the immune system and stimulate the repair in lung microenvironment after SARS-CoV-2 infection by immunomodulatory mechanisms, even in older patients [53]. MSCs secrete multiple paracrine factors (growth factors, cytokines) and their intravenous infusion usually lead them to concentrate in the lungs [54]. Pulmonary functions may be repaired and improved by these factors as they play a significant role in counteracting fibrosis and protecting or rejuvenating alveolar epithelial cells [55]. Older COVID-19 patients would particularly benefit from the use of MSC therapy, considering that these individuals are highly susceptible to SARS-CoV-2-induced pneumonia that results in severe respiratory distress and death because of immunosenescence [56].

Through MSCs therapy, the inhibition of the overaction of the immune system can occur, as well as promoting endogenous repair, improving the microenvironment and the homeostasis. It can prevent the cytokine release syndrome (cytokine storm) caused by abnormally activated immune cells, which deteriorates the patient's state and may cause the disabled function of endothelial cells, capillary leakage, mucus block in lung, and finally respiratory failure, leading to multiple organ failure [5,57].

The regulation and modulation of inflammatory responses together with the promotion of tissue repair may explain the improved outcome of COVID-19 patients after the transplantation of MSCs [53]. MSCs can be an effective therapy for SARS-CoV-2 and other respiratory diseases by decreasing their effects and restoring respiratory function. Further investigations should be made to evaluate this hypothesis.

## 6. Immunotherapy Based in NK Cells

Natural killer (NK) cells are cytolytic innate immune cells originated from the bone marrow and represent up to 10–15% of peripheral blood mononuclear cells [58]. In normal conditions, NK cells are fast-response cells of the innate immune system that strike viruses and eliminate cancer cells [59], through inflammatory signals, such as cytokines or chemokines. By activating surface receptors' engagement, the NK cell recognizes and kills infected cells; additionally, they influence other immune cells, including T cells, to remove the infected cells from circulation [60].

The identification of target cells depends on a balance between NK inhibitory and activating receptors expressed on their surface, which interact with ligands in compromised cells (e.g., infected by viruses) [61]. The function of human NK cells is mainly regulated by Killer Ig-like Receptors (KIRs), and NKG2A inhibitory receptors that interact with class I

auto-HLA molecules. These receptors represent the main verification points for NK cell activation, although other non-specific HLA class I inhibitory receptors have been identified in these cells that also affect the final balance regulating the cytotoxicity of NK cells [58,62]. These include the PVR-related Ig domain (PVRIG, known as CD112R), the lymphocyte-3-activating gene (Lag-3), the T cell immunoglobulin and the ITIM domain (TIGIT), the T cell immunoglobulin and the mucin domain containing 3 (Tim-3), programmed death-1 (PD-1), and probably TACTILE (CD96). Activated CD4+ and CD8+ T cells and a subset of NK cells express Lag-3 [63].

NK cells expressing the activating receptor NKG2C expand specifically in response to human cytomegalovirus (HCMV) [64]. Individuals infected with HCMV had an enhanced response not only to HCMV (the reactivation of HCMV in HSCT) [65], but also to other viruses, including Hantavirus [66], Chikungunya virus [67], hepatitis B and C virus [68], and Herpesviruses [69].

NK cells have been reported to regulate T cell responses through multiple direct and indirect mechanisms, including NK cell-mediated death of activated CD8+ T cells, CD4+ T cells [70], and DCs [71,72]. The activation of NK-mediated DC cells contributes to the development of a potent innate immunity. DC activation, maturation, and cytokine production are provided by signals from NK, promoting adaptive immunity (Figure 3). NK cells express FasL and TRAIL on the membrane surface. These two receptors bind to Fas (CD95) and TRAILR (DR4 and DR5), respectively. The involvement of FasL-Fas/TRAIL-TRAILR resulted in the induction of apoptosis in cells infected by viruses or in tumor cells [73]. Antibody-dependent cell cytotoxicity (ADCC) can be exerted by NK cells, when target cells are coated with IgG antibodies that bind the Fcγ CD16 receptor on NK cells, canceling out inhibitory signals, and triggering cytotoxicity and cytokine secretion [74].

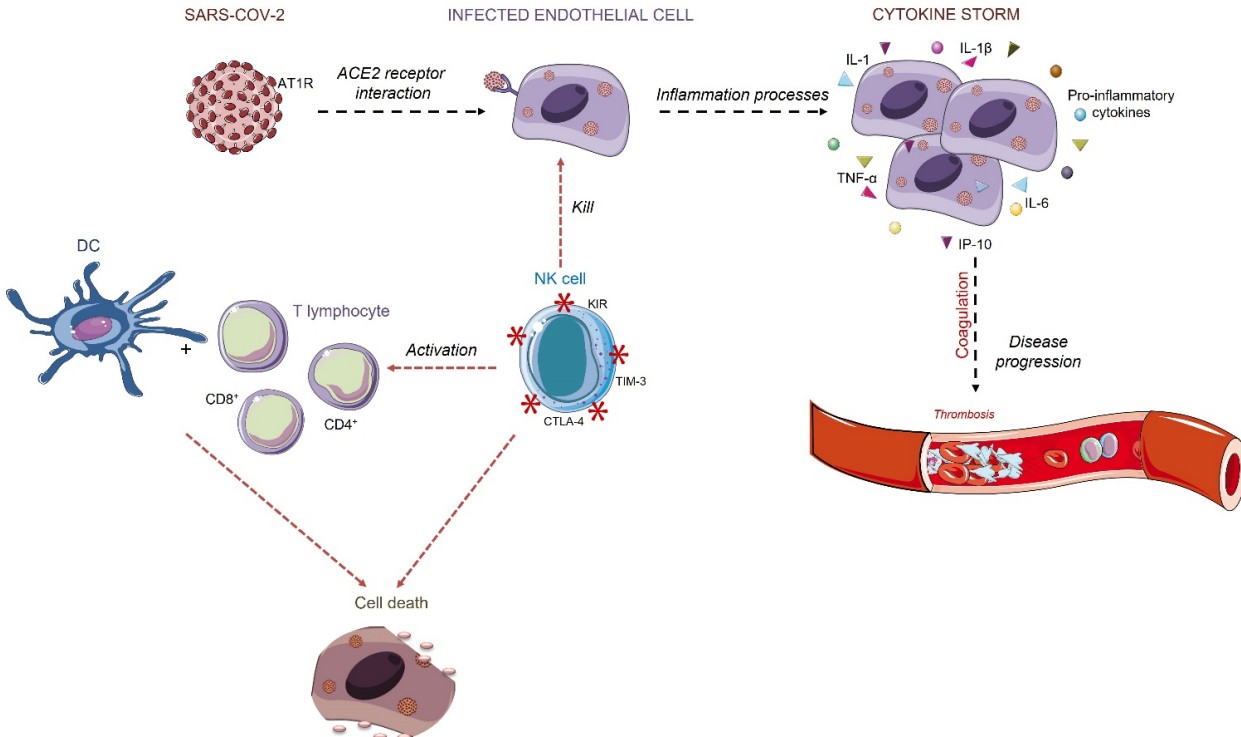

**Figure 3.** Endothelial cells' acute inflammation is caused as a response to the virus infection. The infection may cause intense pro-inflammatory cytokines release, such as IL-1, IL-6, IL-1β, TNF-α, IP-10. The overreaction of the immune system may increase the quantity of the cytokines that contributes to altered coagulation. Intense fibrin formation and its intravascular deposition characterize the inflammation-induced coagulation, creating sites of thrombosis. The infected cells are recognized by NK cells, which activate the cytotoxic receptors to eliminate the infected cell and then activate DCs and T cells response. Triangles and smooth spheres symbols refer to soluble cytokines and growth factors; asterisks refer to cell membrane proteins.

In this way, we argue that NK–T cell interaction exhibits an essential role in both acute and chronic viral infections. NK cells have the potential to shape T cell responses early during activation, primarily by reducing T cell responses and humoral immunity. This shows that NK cells present a regulatory function in adaptive immune responses in the setting of persistent viral infections. Thus, the use of NK cells as an immunotherapy to treat patients with SARS-CoV-2 might open new perspectives on the development of specific therapies, with an important clinical impact in the future.

## 7. Conclusions

Since SARS-CoV-2 emerged at the end of 2019, populations of all over the world have been directly affected by COVID-19 disease or indirectly by essential preventive measures that governments implemented, such as quarantine and community isolation. Although SARS-CoV-2 infectious mechanisms are still being elucidated, some key factors rise as possible therapeutic targets, including TMPRSS2, viral S protein, and Mpro for the development of specific drugs.

Additionally, besides molecular-specific targets, cell therapies based on MSC treatments and NK cells might be effective approaches against SARS-CoV-2 and its sequels. Many COVID-19 patients undergo an aggressive inflammatory storm, which seems to be one of the main reasons for clinical worsening, especially related to acute respiratory distress, one of the main clinical features of this new disease. In this scenario, both MSCs and NK cells have recognized immunomodulatory effects that could be applied as adjuvant treatments against the aggressive pro-inflammatory feature of COVID-19. Cell therapy may be used as a new approach for this new disease and for similar ones in the future. Moreover, it can be a more effective therapy, mainly if associated with other types of treatment, such as anticoagulants or anti-inflammatory drugs. Future studies and clinical trials should evaluate the application of this approach, including for other infectious diseases that affect the human population. To date, the continuous development of COVID-19 vaccines is crucial to avoid new contamination and the emergence and spread of new viral variants. Individual and community protection measures against viral infection are still essential, especially in intensely affected regions and countries.

**Author Contributions:** Conceptualization: D.A.D.C., A.S.P. and P.L.D.-S.-J. writing—original draft preparation, review, and editing: D.A.D.C., A.S.P., N.F.L., P.L.D.-S.-J.; supervision: P.L.D.-S.-J. All authors have read and agreed to the published version of the manuscript.

**Funding:** This research received no external funding.

**Data Availability Statement:** No new data were created or analyzed in this study. Data sharing is not applicable to this article.

**Conflicts of Interest:** The authors declare no conflict of interest.

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
