# Peer review of "A COVID-19 Overview and Potential Applications of Cell Therapy"

_biologics, doi:10.3390/biologics1020011_

Round 1

Reviewer 1 Report

Line 28:  such as but not only mass....

should read  
 such as, but not limited to 

Line 42: 

 COVID-19 has leading 43 many people to 

should read

 COVID-19 has resulted in the death of many people, and despite.....

Line 45: 

 measures, the only 

should read

 isolation and masking measures, the only effective non pharmacologic procedure in preserving.....

Line 122 Label figures A, B, C, D to match with legend. (for other figures check this also).

Line 133, mechanism(s) not mechanism. 

Line 149: 

Once humans do not have Mpro close homologues 

should read

If humans do not have Mpro close homologues 

Line 163:  breathing rather than the word breath should be used.

Line 169: 

Patients that show severe and critic ...

should read

Patients that show a severe or critic ....

Line 171: resulting (not resulted)

Line 262:  should read.......of the innate immune system..

 measures which is the only 

Author Response

Thank you very much for considering our manuscript to be published in the Biologics. We also appreciate the important comments and suggestions that the Reviewers have made. We have revised our manuscript and below we present a response for the comments.

We thank the Reviewer for the corrections which were pointed out and we have made all the suggested alterations. In addition, we reviewed again the full text of the article regarding the English language. All the alterations were highlighted in red text in the manuscript archive.

We appreciate the Reviewers’ pertinent suggestions and comments which have helped us to improve this article, and would like to thank them for their cautious reading of the text. The text that has been added or amended in response to the Reviewers’ comments is highlighted in red text in the manuscript archive. We hope that the changes that we have made have fully responded to their comments.
The final version of our manuscript has been read and approved by all the listed authors, who have each provided the attention necessary to ensure the integrity and improvement of the article.

Thank you for your consideration.

Kind regards.

Reviewer 2 Report

This review by Câmara et al. is a well-written compendium of literature data on CoViD-19 and SARS CoV-2 molecular and biochemical features, with an original input represented by the in-depth analysis of the possible role of MSCs as a new tool for the disease therapy alone, or together with traditional drugs.

In my opinion, this review constitutes a good scientific contribution.

I only suggest a revision of the text and figure 3 for few typos and grammatical errors. Furthermore, since the Authors reported in the figure legends (Fig. 2 and 3) the description of panels a, b, etc., I would insert the corresponding letters in the image, for sake of clarity and completeness. 

Author Response

Thank you very much for considering our manuscript to be published in the Biologics. We also appreciate the important comments and suggestions that the Reviewers have made. We have revised our manuscript and below we present a response for the comments.

We appreciate the comments made by the Reviewer. We have corrected the Figures and added the description (letters) in each panel. All the figures were updated in the manuscript archive and were resubmitted into the Biologics system. We also reviewed the full text in order to correct any typo or grammatic errors. 

We appreciate the Reviewers’ pertinent suggestions and comments which have helped us to improve this article, and would like to thank them for their cautious reading of the text. The text that has been added or amended in response to the Reviewers’ comments is highlighted in red text in the manuscript archive. We hope that the changes that we have made have fully responded to their comments.
The final version of our manuscript has been read and approved by all the listed authors, who have each provided the attention necessary to ensure the integrity and improvement of the article.

Thank you for your consideration.

Kind regards.
